# SEMI-LOCAL SEARCH FOR LR SCHEDULES

## ABSTRACT

The learning rate schedule is a critical parameter of the optimization pipeline in modern machine learning. Unfortunately, searching for the optimal schedule is very difficult because the simple "local search" method of using the learning rate that does best on the very next iteration performs poorly in practice: industry standard schedules such as cosine decay or WSD trade-off worse early performance for better final performance. We investigate the extent to which a "semi-local" search that only looks a few iterations ahead can rectify this problem in order to design an automated procedure to search for good learning rate schedules. Our experiments rigorously establish that simple greedy search methods fail to find optimal schedules, but that a limited amount of non-locality in the search *can* design better schedules.

## 1 INTRODUCTION

Most stochastic optimization algorithms update iterates $x_t$ in the following form:

$$x_{t+1} = x_t + \eta_t \Delta_t, \tag{1}$$

where $\Delta_t$ is the update tensor (e.g., SGD uses the stochastic gradient at $x_t$ as the update), and $\eta_t$ is the learning rate (LR) at step $t$. Designing a good update $\Delta_t$ is important and has received most of the attention in the literature, but choosing a proper sequence of $\eta_t$ (known as the learning rate schedule) is also very important.

Searching for the globally optimal LR schedule is practically infeasible because the space of all schedules is extremely high dimensional and it is unclear what structure may be present in this space to aid any global searach. Standard theoretical analyses demonstrate that different LR schedules, such as $\eta_t \propto 1/t, \eta_t \propto 1/\sqrt{t}, \eta_t \equiv \eta$, perform well under various assumptions, but their practical performance is worse than typical empirically-discovered schedules Defazio et al. (2023).

The standard practical way to set the schedule is to search among a few pre-defined sub-families of schedules, such as the step decay schedule (Krizhevsky et al., 2012), cosine annealing schedule (Loshchilov & Hutter, 2017), and the warmup-stable-decay (WSD) schedule (Hu et al., 2024). These schedules are parametric, meaning they are defined by a few parameters such as the base learning rate, warmup and decay step portion (WSD schedule), and decay period (step decay schedule). Consequently, an optimal schedule can be found via grid search on these parameters.

Alternatively, there have been prior works that aim to learn learning rate schedules on-the-fly (Chen et al., 2017; Li et al., 2017; Antoniou et al., 2018). The essential idea is applying some form of "gradient-descent on the learning" in order to learn the locally optimal learning rate in every step. While this idea is tempting, this is not popular in practice, and our results confirm the hypothesis that such schemes (which are an inherently local search) are likely ineffective. However, this does not mean that we should completely give-up on attempting to learn a learning rate schedule: it just means that the simple local objective is not correct. Instead, it raises an important question: is there any way to learn a learning rate schedule that matches or even surpasses practical schedules?

We answer this question affirmatively by introducing a new algorithm, "semi-local search", which learns an effective learning rate schedule on-the-fly. At a high level, this algorithm selects suboptimal learning rates instead of greedily choosing locally optimal learning rates (hence the name *semi-local search*). Surprisingly, adding a small amount of semi-locality significantly improves the quality of the resulting schedule. This finding is summarized in Figure 1 and Table 1. Although we do not explore this possibility here, our positive results for semi-local search suggest that it may

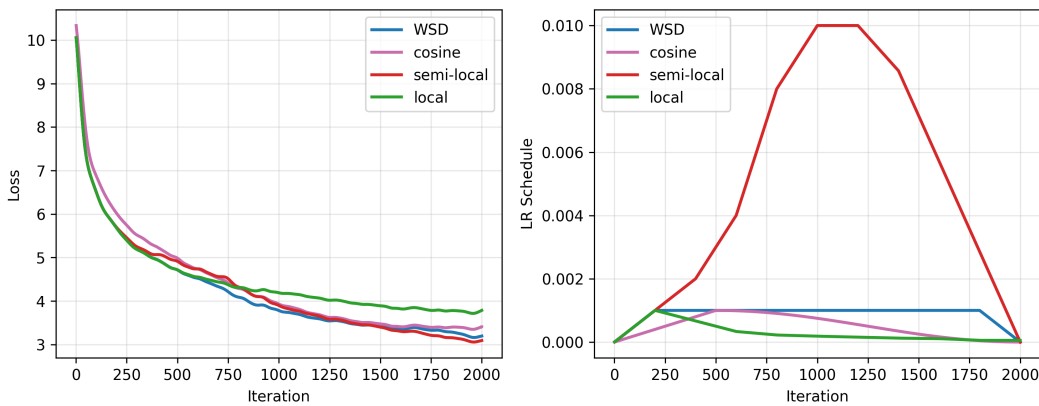

Figure 1: Train loss (left) and LR schedule (right) for local search, semi-local search, and state-of-the-art baselines, each carefully tuned through grid search. The semi-local search algorithm is introduced in Section 3, and experiment details including tuning methods are provided in Section 4. Local search favors smaller learning rates and underperforms compared to the baselines, while semi-local search explores larger learning rates more aggressively, trading sub-optimal early performance for better end performance that exceeds the baselines.

Table 1: Train loss (top) and eval loss (bottom) of local search, semi-local search, and baselines.

|            | WSD   | Cosine | Local | **Semi-Local** |
|------------|-------|--------|-------|------------|
| Train loss | 3.184 | 3.438  | 3.761 | **3.093**  |
| Eval loss  | 3.745 | 3.995  | 4.401 | **3.612**  |

indeed be possible to perform some kind of regularization to rescue a local search procedure. That is, although our method incurs a nontrivial (although still manageable) computational burden, our work points the way towards a future design of more efficient methods for designing schedules.

To summarize the empirical contributions of our paper:

- We introduce *semi-local search*, a new search method that learns effective learning rate schedules and is capable of recovering most of the practical schedules.
- We empirically observe that local search underperforms compared to state-of-the-art baselines, whereas semi-local search surpasses them.
- We propose new schedules with closed form inspired by semi-local search, and we demonstrate that they are better than the baseline schedules.

## 2  FURTHER RELATED WORK

There have been a long history of searching for effective learning schedules. Early results in optimization theory have shown that either constant $\eta_t \equiv \eta$ or $\eta_t \propto 1/\sqrt{t}$ is optimal for convex or smooth non-convex optimization problems (Zinkevich, 2003; Ghadimi & Lan, 2013); and $\eta_t \propto 1/t$ is optimal (Rakhlin et al., 2012; Shamir & Zhang, 2013) for strongly convex problems.

Schedules popular in practice look rather different. For training ConvNets, the classical choice was the stepwise decay schedule, which starts at a constant learning rate and decays periodically (Krizhevsky et al., 2012; He et al., 2016). More recent empirical work on Transformer models has employed cosine schedules Loshchilov & Hutter (2017); Hoffmann et al. (2022), and (even more recently) trapezoidal-shaped "warmup-stable-decay" (WSD) schedules (Hu et al., 2024). These schedules essentially always outperform the classical schedules (often quite dramatically, and even on convex objectives (Defazio et al., 2023)).

From a theoretical side, recent work has established that, at least for convex objectives, a simple linear decay schedule is also worst-case optimal (Zamani & Glineur, 2023; Defazio et al., 2023), and in certain regimes this theory can provide insight into the empirically-motivated WSD or cosine schedules (Schaipp et al., 2025).

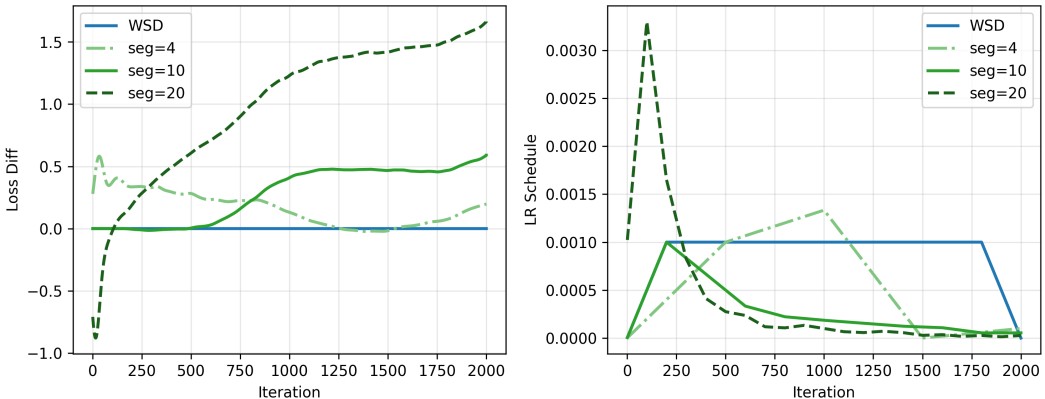

Figure 2: Local search with different number of total segments. Left: loss difference computed as (*local search−baseline*), where the zero line is a carefully tuned WSD baseline; Right: LR schedule. Every local search prefers relatively small LRs and underperforms the baseline (positive loss diff). Moreover, the performance degrades as the number of segments increases.

## 3 METHOD

### 3.1 LOCAL SEARCH IS SUBOPTIMAL

The local search method can be summarized as follows: Given $T$ total training steps, we split them into $K$ segments, and in each segment a locally optimal schedule is chosen which minimizes the loss in that segment. This procedure is formally defined in the pseudo-code Algorithm 1 (with $\epsilon = 0$) which will be discussed in details in the next section.

Although the idea of local search is intuitively appealing, our experiment results show that its empirical performance does not match the state-of-the-art schedules (we will defer the experiment setup in Section 4). Figure 2 reports the performance of local search methods with 4, 10, and 20 segments, compared to well-tuned state-of-the-art baselines. The results show that local search is worse to the baselines, and its performance degrades as the number of segments increases. As the number of segments grows, the search method becomes increasingly local—in the extreme case where the number of segments equals the number of iterations, local search greedily chooses the learning rate that minimizes the loss at each step. Consequently, this observation suggests that finding a good schedule requires decreasing locality.

Figure 2 further shows the evolution of training loss and the LR schedule. We observe that local search (especially with more segments) prefers smaller learning rate compared to baselines, and smaller learning rates correspond to slower convergence of the training loss. Combining these observations, we hypothesize that the following conditions helps designing an effective LR search method: (1) the search method should not greedily choose the locally optimal learning rate, and (2) under certain constraint, a larger learning rate should be encouraged. This leads to the design of the semi-local search in the next section.

### 3.2 SEMI-LOCAL SEARCH

We propose the following method, called semi-local learning rate search. Given $T$ total training steps, we split them into $K$ segments, and in each segment a sub-optimal schedule is chosen. Unlike local search, which greedily selects the optimal schedule at every step, our method intentionally selects a sub-optimal schedule, hence the name *semi-local* search. This procedure is formally defined in the pseudo-code Algorithm 1 and illustrated in Figure 3.

This algorithm finds a continuous piecewise linear schedule. In the extreme case when the number of segments is $T$ and the set of learning rate candidates is infinite, semi-local search is capable of representing any learning rate schedule. Although this setup is practically infeasible, we design the learning rate candidates such that the algorithm is still able to represent many well-known schedules.

The constant $\epsilon_k$ is a quantitative measure of semi-locality. By setting $\epsilon_k \equiv 0$, this method recovers the local search—finding the optimal learning rate that leads to the lowest local loss in every

**Algorithm 1** Semi-Local LR Search

1: **Input:** Segments $(\tau_0 = 0) \leq \tau_1 \ldots \leq (\tau_K = T)$, learning rate candidates $\eta_k^{(i)} \geq 0$, constants $\epsilon_k \geq 0$, initial LR $\eta_0 = 0$, and initial checkpoint $C_0$.
2: **for** $k = 1$ **to** $K$ **do**
3:     Submit $n$ parallel jobs in the $k$-th segment $[\tau_{k-1}, \tau_k]$:
4:        Job $i$ resumes training from $C_{k-1}$ with a linear schedule from $\eta_{k-1}$ to $\eta_k^{(i)}$.
5:        Job $i$ returns checkpoint $C_k^{(i)}$ and averaged loss $\ell_k^{(i)}$.
6:     Choose $i^*$ such that

$$i^* = \underset{i \in [n]}{\arg\max} \left\{ \eta_k^{(i)} : \ell_k^{(i)} \leq \min_j \ell_k^{(j)} + \epsilon_k \right\}. \tag{2}$$

7:     Update $\eta_k = \eta_k^{(i^*)}$ and $C_k = C_k^{(i^*)}$.
8: **end for**

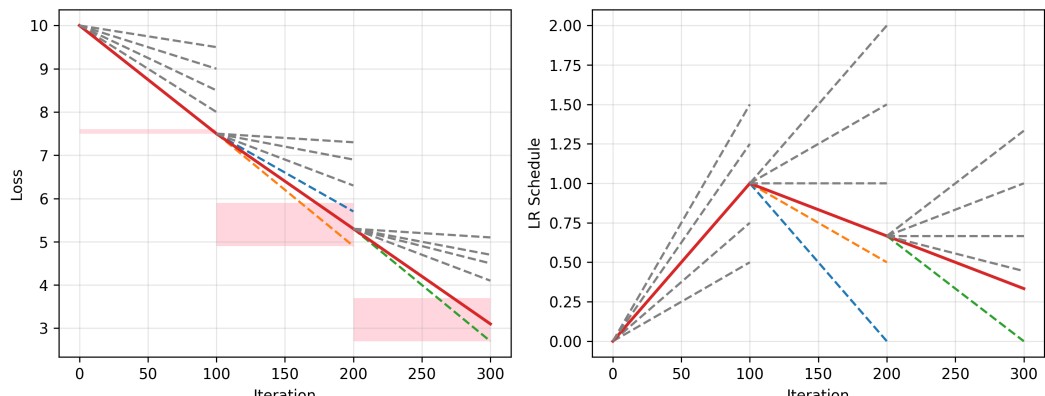

Figure 3: Illustration of the semi-local search algorithm: In each segment, the algorithm selects a sub-optimal run $i^*$ (shown as the bold red line) according to equation 2. Specifically, among all runs whose losses lie within $\epsilon_k$ of the lowest loss (this range is indicated by the pink-shaded area, with all runs within it colored), $i^*$ is chosen as the run with the largest learning rate. The corresponding learning rate is then used as the starting point of the linear schedules in the next segment, and the associated checkpoint (which includes both model weights and the optimizer internal state, ensuring that the optimizer is not restarted) serves as the checkpoint for continuation.

segment, which is shown sub-optimal in the previous section. On the other hand, as $\epsilon_k \to \infty$, the semi-local search always picks the largest learning rate in the candidate set, namely $\eta_k = \max_i \eta_k^{(i)}$, which also performs poorly if that learning rate is too large. As a result, a proper amount of semi-locality is needed. We elaborate these details in the following.

### 3.2.1 IMPLEMENTATION DETAILS

The semi-local search method has two critical inputs, the learning rate candidates $\{\eta_k^{(i)}\}_{i=1}^n$ and the semi-locality constant $\epsilon_k$. As aforementioned, a larger candidate set improves the representability of the algorithm, but increases the computation cost at the same time. Meanwhile, a proper semi-locality constant is needed to interpolate between greedily choosing locally optimal learning rates and blindly choosing the largest learning rates.

To achieve these goals, we set the learning rate candidates in the $k$-th segment to be the last learning rate $\eta_{k-1}$ times a fixed grid defined as $\{\frac{0}{1}, \frac{1}{2}, \frac{2}{3}, \ldots, \frac{n-1}{n}, 1, \frac{n}{n-1}, \ldots, \frac{3}{2}, \frac{2}{1}\}$. Notably, this grid recovers most of the popular schedules. For example, setting $\eta_k = \eta_{k-1}$ recovers constant schedule; setting $\eta_k = \frac{K-k}{K-k+1} \cdot \eta_{k-1}$ recovers linear decay and $\eta_k = \frac{K-k+1}{K-k} \cdot \eta_{k-1}$ recovers linear increase; setting $\eta_k = \alpha \cdot \eta_{k-1}$ for some $\alpha$ in the grid recovers geometric decay/increase. Combining these schedules recovers many more schedules, such as the linear decay with warm-up schedule and the WSD schedule. Figure 3 shows an illustrative example that semi-local search is able to recover the linear decay with warm-up schedule.

Although larger $n$ improves representability, the marginal gain diminishes as $n$ increases. For example, when the grid size is increased from $n = 1000$ to $n = 1001$, the resulting schedules remain close to the existing ones; in contrast, increasing the grid size from $n = 2$ to $n = 3$ introduces many new schedules that differ significantly from those already present. Considering the tradeoff between computation cost and utility, we choose $n = 10$ for the learning rate candidates.

Choosing a proper semi-locality constant $\epsilon_k$ is difficult. Since we do not know how $\epsilon_k$ quantitatively affects the training performance, we manually tune it in order to find a nearly optimal value. In most experiments, we fix $\epsilon_k \equiv \epsilon$ and tune $\epsilon$ in a geometric grid. More specifically, we start with $\epsilon = 0.06$, an arbitrary value chosen to be the initial guess, and run semi-local search with $\epsilon, 0.5\epsilon, 2\epsilon$. If the performance of $2\epsilon$ is better than that of $\epsilon$, we double $\epsilon' \leftarrow 2\epsilon$, test semi-local search with $2\epsilon'$, and compare $\epsilon'$ vs $2\epsilon'$. We repeat this process until the performance regresses. Similarly, we halve $\epsilon$ if $0.5\epsilon$ is better than $\epsilon$. Besides constant $\epsilon_k \equiv \epsilon$, we also test linearly decaying $\epsilon_k = \epsilon \cdot \frac{K-k}{K-1}$. Investigating more advanced sequence of $\epsilon_k$, such as adaptively choosing $\epsilon_k$ on-the-fly, is not the main focus of our paper, and we leave that for future work.

There are three exceptions of choosing learning rate candidates and semi-locality constant. The first two exceptions are the first and the last segments. In the first segment, we fix $\epsilon_1 = 0$ and set the learning rate candidate to be a multiplicative grid $\{1e{-}1, 3.33e{-}2, 1e{-}3, \ldots, 1e{-}5\}$. Since the initial learning rate and checkpoint provides no meaningful information of training, we use the standard grid search to find a proper learning rate to start with. In the last segment, we set $\epsilon_K = 0$ because training is complete and the locally optimal learning rate should be used in this segment. All intermediate segments follow the aforementioned setup. The third exception occurs if the starting learning rate of some intermediate segment $k$ is 0. In this case, we again apply the same grid as the first segment and set $\epsilon_k = 0$ in order to determine the next learning rate via standard grid search.

# 4 EXPERIMENT RESULTS

We evaluate the proposed semi-local search method on language model pre-training task. In particular, we pre-train a variant of 124M minGPT model (Karpathy, 2020) on the pile dataset (Gao et al., 2020) using the AdamW optimizer (Loshchilov & Hutter, 2018). Due to limited resources, in most experiments we train the model with 2000 total steps, corresponding to 0.26B training tokens, and evaluate with 0.22B evaluation tokens. We also run several scaled-up experiments, in which the total training steps is increased to 10000 and the training tokens is increased to 1.3B. For reproducibility, all sources of randomness, including model initialization and data streaming order, are fixed across all experiments. More details are discussed in A.

## 4.1 FINE-TUNING BASELINE

The semi-local search method is implemented following the description in 3.2.1. To compare its performance with state-of-the-art schedules, we choose the cosine annealing schedule and the WSD schedule as baselines. We fine-tuned the baselines via grid search over all the parameters including base learning rate, warmup and decay steps. See Appendix B.1 for detailed tuning results.

For the cosine schedule with 2000 steps, we first tuned the base learning rate, searching over learning rates in the grid $\{1e{-}2, 3.33e{-}2, 1e{-}3, 3.33e{-}3, 1e{-}4, 3.33e{-}4, 1e{-}5, 3.33e{-}5\}$, and fixed warmup steps to 200, a standard choice of 10% of the total step. Then, we tuned the warmup steps while fixing the baseline learning to 1e-3, the optimal value found in the previous step. We searched over the grid $\{0\%, 5\%, 10\%, 25\%, 50\%\}$ of the total steps, and found that 25% has the optimal eval loss 3.995.

We followed a similar procedure for the WSD schedule, fixing 200 warmup and decay steps when tuning the base learning rate. Then we tuned warmup steps using the optimal base learning rate and 200 decay steps, and finally we tuned decay steps using the optimal warmup step. As a result of tuning, the best WSD schedule has base learning rate 1e-3, 200 warmup steps and 200 decay steps, achieving eval loss **3.745**. In addition, we also tuned a variant of the WSD schedule, linear decay with warmup (i.e., WSD without the "stable" phase). We fixed base learning rate to 1e-3 and tuned the warmup steps over the grid $\{0\%, 5\%, 10\%, 25\%, 50\%\}$, and we found that 20% has the best eval loss 3.955. For the scaled-up experiments with 10000 total steps, we tuned the baseline learning rate of the WSD schedule while fixing warmup and decay steps to 1000, 10% of total steps. The optimal learning rate is 3.33e-4 with eval loss 3.261.

Table 2: All-in-one table of train loss (top) and eval loss (bottom) of all experiments in the figures.

| | **WSD** | 4 segs | 10 segs | 20 segs | | | | |
|---|---|---|---|---|---|---|---|---|
| Fig. 2 | **3.184** | 3.387 | 3.761 | 4.869 | | | | |
| | **3.745** | 3.979 | 4.401 | 5.432 | | | | |
| | WSD | $\epsilon = 0.0$ | 0.06 | **0.12** | 0.24 | 0.48 | | |
| Fig. 4 | 3.184 | 3.761 | 3.165 | **3.123** | 3.160 | 3.136 | | |
| | 3.745 | 4.401 | 3.711 | **3.653** | 3.683 | 3.658 | | |
| | WSD | const | **decay** | | | | | |
| Fig. 5 | 3.184 | 3.123 | **3.093** | | | | | |
| | 3.745 | 3.653 | **3.612** | | | | | |
| | WSD | 4 segs | **10 segs** | 20 segs | | | | |
| Fig. 6 | 3.184 | 3.158 | **3.123** | 3.653 | | | | |
| | 3.745 | 3.697 | **3.653** | 4.173 | | | | |
| | WSD | $\epsilon = 0.0$ | 0.015 | 0.03 | 0.06 | 0.12 | 0.24 | **0.48** |
| Fig. 7 | 3.184 | 4.869 | 4.346 | 4.190 | 4.309 | 4.918 | 5.143 | **3.653** |
| | 3.745 | 5.432 | 4.918 | 4.779 | 4.879 | 5.453 | 5.680 | **4.173** |
| | WSD | stretched | $\epsilon = 0.12$ | $\epsilon = 0.24$ | | | | |
| Fig. 8 | 2.826 | 2.920 | 2.677 | **2.648** | | | | |
| | 3.261 | 3.433 | 3.108 | **3.070** | | | | |
| | WSD | **semi-local** | quadratic | triangle | | | | |
| Fig. 9 | 3.184 | **3.093** | 3.117 | 3.025 | | | | |
| | 3.745 | **3.612** | 3.632 | 3.652 | | | | |

## 4.2 Semi-Local Search

In this section, we present the main observations of semi-local search. We choose the optimally tuned WSD schedule from the previous section as the baseline. In all graphs, we report the difference of the train loss of semi-local searches minus that of the baseline in the left plot, and we include the learning rate schedule in the right plot. We also report the train loss and eval loss of all experiments in an all-in-one table, see Table 2. Details of all experiments not included in the main text can be found in B. The loss differences in the plots are smoothed with a Gaussian filter of standard deviation 10, and the numerical training losses reported in the table correspond to the final values of the smoothed training losses.

**Semi-locality constant** First we study how semi-locality constant affects the performance of semi-local search, reported in Figure 4. We use 10 total segments and 2000 train steps, and we fix $\epsilon_k \equiv \epsilon$ for $1 < k < K$ and $\epsilon_1, \epsilon_K = 0$. As discussed in Section 3.2.1, we tune $\epsilon$ in the following way: we start with $\epsilon = 0.06$ as the initial guess and repeatedly double it until the performance regresses at $\epsilon = 0.48$. The optimal semi-locality constant is $\epsilon = 0.12$ and achieves eval loss 3.653, which outperforms the carefully tuned WSD baseline with eval loss 3.745.

Notably, while local search exhibits a substantial loss gap ($\approx 0.6$), introducing even a small degree of semi-locality (e.g., $\epsilon = 0.06$) yields immediate gains and surpasses the baseline. This finding challenges the underlying intuition of prior "hyper-gradient descent" methods and their variants, which essentially finds locally optimal learning rates in every step. At the same time, it suggests a new avenue for designing learning-rate schedules, either closed-form or adaptively learned on-the-fly, that leverages semi-locality.

**Should semi-locality decay over time?** While adding semi-locality improves performance, the proper degree of semi-locality remains quantitatively uncertain. Even if grid search provides a tuned value, an important question persists—should semi-locality remain fixed over time, or should it vary as training progresses? As we ran previous experiments, we observed that the standard deviation of the losses across parallel jobs decreases in later segments, suggesting a possibility that the required semi-locality might decrease over time. To verify this hypothesis, we proposed a linearly decay policy on semi-locality constants, formally defined as $\epsilon_k = \epsilon \cdot \frac{K-k}{K-1}$ for $k > 1$ and $\epsilon_1 = 0$. We use 10 total segments and 2000 train steps, and we tune $\epsilon$ in a similar way. Since $\epsilon_k$ decreases over time, we picked a much larger value $\epsilon = 0.24$ as the initial guess and doubles it until performance

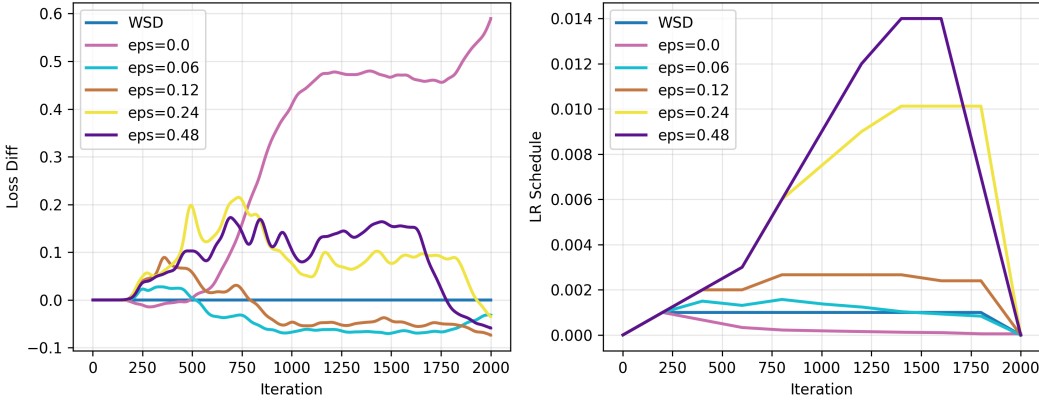

Figure 4: Semi-local search with 10 segments and 2000 total train steps, using different semi-locality constant $\epsilon$. Left: loss difference computed as (*semi-local − baseline*), where the zero line is the WSD baseline; Right: LR schedule. $\epsilon = 0$ (local search) results in small LRs and underperforms, whereas moderate $\epsilon$ (0.06, 0.12) chooses slightly larger LRs and yields consistently lower loss than the baseline (negative diff). A larger $\epsilon$ (0.24, 0.48) explores aggressive LRs in the early stage, incurring sub-optimal performance mid-training but better performance at the end. Overall, semi-local search outperforms the well-tuned WSD schedule, and a larger $\epsilon$ trades early performance for good end performance.

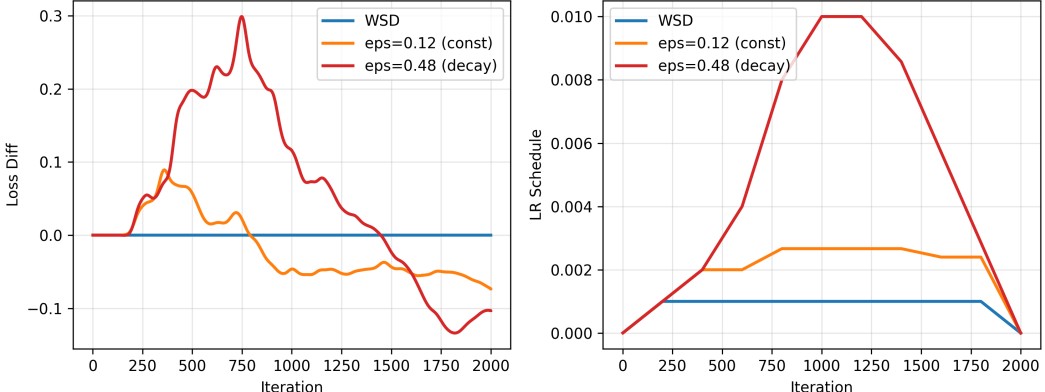

Figure 5: Semi-local search with 10 segments and 2000 total train steps, using constant $\epsilon_k \equiv \epsilon$ (orange) and linear decay $\epsilon_k = \epsilon \cdot \frac{K-k}{K-1}$ (red) where $\epsilon$ is carefully tuned. Left: loss difference w.r.t. the WSD baseline (zero line); Right: LR schedule. The optimal $\epsilon$ of linear decay policy is much larger compared to the constant policy (0.48 vs 0.12). Moreover, the decay policy allows a more aggressive LR exploration, reaching a larger peak LR compared to the constant policy (0.01 vs roughly 2e-3). Notably, the aggressive LR exploration phase (first 750 steps) correlates with an increasing loss difference, which then decreases steadily and ultimately yields better final performance.

degrades at $\epsilon = 0.96$. Under this policy, $\epsilon = 0.48$ yields the best eval loss 3.612, outperforming the best constant-policy semi-local search (3.653). See Figure 5 and Table 2, row 2. These results confirm our hypothesis, showing that decaying semi-locality can outperform a fixed setting for $\epsilon$.

**Number of segments** Another important variable of semi-local search, besides semi-locality constant, is the number of total segments. Besides 10 segments, we also carefully tune the algorithm with 4 segments and 20 segments. See Figure 6 and Table 2, row 3. We observe an apparent performance degradation with 20 segments, and to better understand the underlying reason, we peak into the performance of schedules with 20 segments and different semi-locality, shown in Figure 7. In particular, we notice that the schedule with $\epsilon = 0.12, 0.24$ decreases to 0 at step 600, implying that all other learning rate candidates are at least $\epsilon$ worse compared to LR=0. This suggests that we need a finer learning rate candidate grid when number of segments increases.

**Performance under larger scale** We have demonstrated that semi-local search shows better performance compared to the WSD baselines with 2000 train steps. We further verify its performance on a larger scale with 10000 train steps, and we tune semi-local search with 10 segments and con-

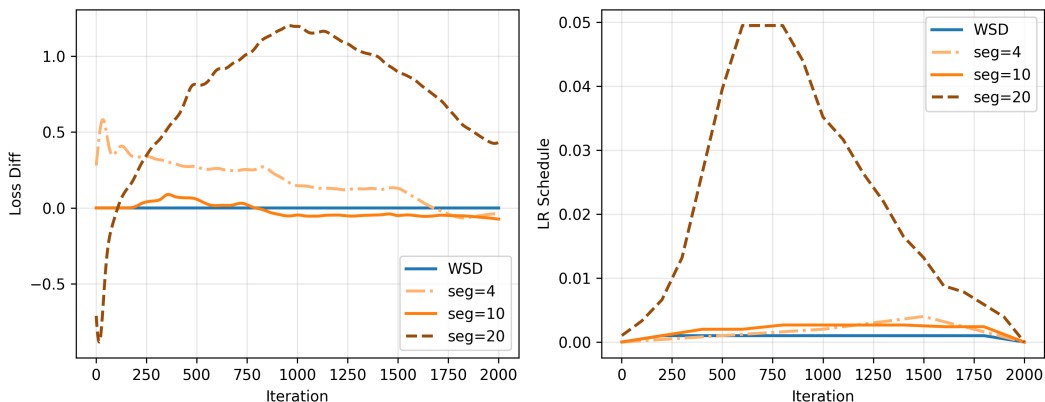

Figure 6: Semi-local search with 2000 train steps and carefully tuned $\epsilon_k \equiv \epsilon$, using different number of total segments. Left: loss difference w.r.t. the WSD baseline (zero line); Right: LR schedule. 4 segments and 10 segments yields similar performance; with 20 segments the schedule reaches a much larger peak values and its performance regresses significantly.

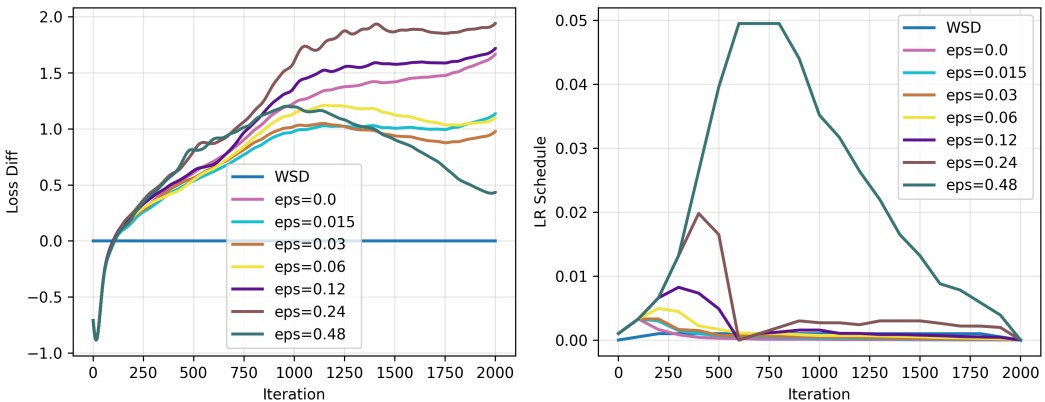

Figure 7: Semi-local search with 20 segments and 2000 training steps, using different semi-locality constants $\epsilon$. Left: loss difference w.r.t. the WSD baseline (zero line). Right: LR schedules.

stant policy $\epsilon_k \equiv \epsilon$. For comparison, we choose the best semi-local schedule from 2000 steps (10 segments, $\epsilon = 0.48$ with linear decay policy) and stretch it to 10000 steps. We then re-tune the peak learning rate via grid search. The result is reported in Figure 8 and Table 2, row 4.

The stretched schedule has the same peak LR as the WSD baseline, but it takes a longer warmup phase reaching to the peak, resulting in significant loss gap early on and slightly worse end performance. On the other hand, semi-local search from scratch explores much larger peak LR (roughly 3e-3 and 5e-3 respectively vs 3.33e-4). Unlike previous observation that aggressive LR often correlates with degraded early performance, semi-local search shows significantly negative loss diff since the beginning and this negative gap is maintained until the end.

### 4.3 NEW SCHEDULES INSPIRED BY SEMI-LOCAL SEARCH

The best tuned semi-local search ($\epsilon = 0.48$ with linear decay policy shown in Figure 5) inspires the design of new closed-form schedules. In particular, this schedule exhibits a long learning-rate warm-up phase (taking nearly half of the total training steps) and reaches a much larger peak learning rate compared to the best-tuned WSD baseline, followed by a decay phase. Motivated by these features, we study two schedules: a symmetric quadratic schedule (downward parabola) and a symmetric linear schedule (triangle). Unlike the WSD schedule and the cosine decay with linear warm-up, these schedules are parameterized only by the peak learning rate once the number of training steps is fixed. We tune the peak learning rate via grid search and compare the best-tuned variants against the WSD baseline and the best semi-local schedule (see Figure 9).

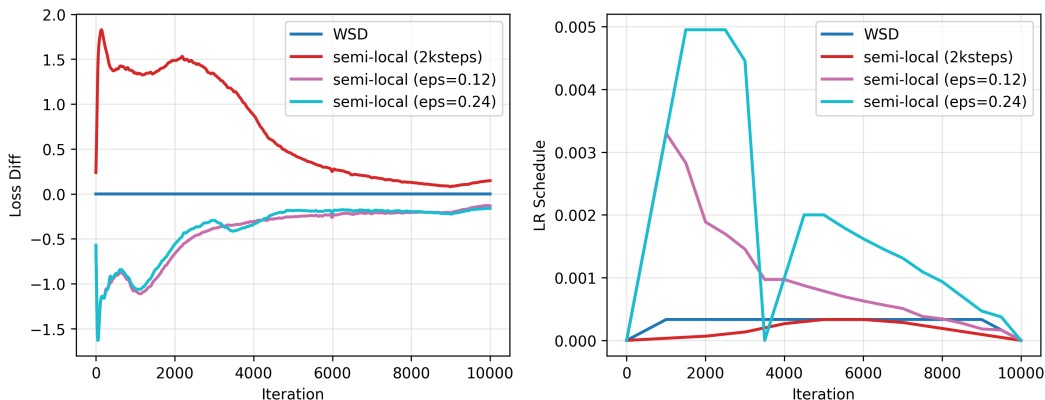

Figure 8: Semi-local search with 10000 train steps. Left: loss difference w.r.t. the WSD baseline (zero line); Right: LR schedule. The "2ksteps" curve (red) is the optimal 2000-step semi-local schedule stretched to 10000 steps and re-tuned via grid search; the other two curves are semi-local search with 10000 steps trained from scratch with $\epsilon = 0.12, 0.24$. Semi-local search from scratch still outperforms the WSD baseline under scaled-up setup, whereas simply stretching and re-tuning the optimal 2000 steps semi-local schedule yields sub-optimal performance.

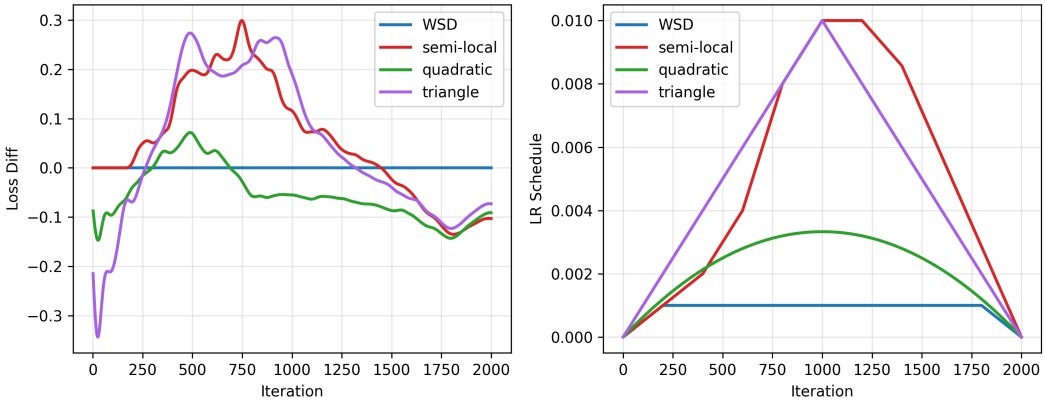

Figure 9: Closed-form schedules inspired from semi-local search: a symmetric quadratic (parabola) and a symmetric linear (triangle) schedule. Left: loss difference relative to WSD (zero line). Right: LR schedule. The triangle has a similar shape to the schedule resulted from the optimal semi-local search, showing early regressions but strong late gains; the quadratic peaks lower ($\approx$3e-3), showing better early performance and matching end performance. Both simple schedules match the performance of the optimal semi-local search and significantly outperforms the WSD baseline.

The best-tuned triangle schedule exhibits both similar shape and performance to the semi-local schedule. More interesting, the quadratic schedule achieves the matching performance, even though its peak learning rate (3.33e-3) is much smaller than semi-local schedule (0.01).

## 5 CONCLUSION

We proposed a new method, semi-local search, for discovering effective learning-rate schedules, showing that it outperforms state-of-the-art baselines, while local search is observed to be significantly less effective. Due to limited resources, our experiments are restricted to relatively small scales, and we hope our ideas can be verified on larger-scale setups. More importantly, our findings open a new avenue for designing effective schedules: our results should be viewed as a proof-of-concept that biasing local methods towards larger learning rates can produce useful schedules. It is our hope that this evidence motivates research into practical automatic schedule design.

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

## A EXPERIMENT DETAILS

**Task**   We pre-train a randomly initialized minGPT model (Karpathy, 2020) with approximately 124M trainable parameters from scratch using the pile dataset (Gao et al., 2020). We use AdamW (Loshchilov & Hutter, 2018) as the base optimizer with momentum constants $b_1 = 0.9, b_2 = 0.999$ and weight decay 0.1 across all experiments. The learning rate is set by the schedule from corresponding experiments.

**Training**   In most experiments, we train the model for 2000 iterations with batch size 128, yielding a total of 0.26B training tokens. For scaled-up experiments, we increase to 10k iterations and correspondingly 1.3B training tokens. We evaluated the trained model on an independent evaluation set with 0.22B tokens.

We fix all sources of randomness including model initialization and data streaming order across all experiments. In particular, this guarantees all experiments, including the parallel jobs submitted by semi-local search, receive the same data batch $\xi_t$ in iteration $t$, hence ensuring a fair comparison.

Moreover, we implement an early stopping policy, which terminates an experiment if either scenario happens: (1) the training loss becomes NaN for more than 3 iterations, or (2) the last training loss is larger than the average loss plus a threshold of 0.5. We resubmit failed runs for a maximum of 3 attempts from the latest checkpoint, and we keep the same data ordering in new attempts. Consequently, we only resubmit failed runs due to either numerical instability or randomness within the CUDA kernel, but not due to data ordering or model initialization.

**Runtime**   All code is implemented using JAX (Bradbury et al., 2018), and all experiments are run on 1×L40 Nvidia GPU under CUDA12.2. To train 2000 iterations, baseline experiments take 3 hours, and local and semi-local search take 12 to 18 hours depending on the number of segments. To train 10k iterations, baselines take 15 hours, and local and semi-local search take 2 to 3 days.

## B ADDITIONAL EXPERIMENT RESULTS

### B.1 FINE-TUNING BASELINE SCHEDULES

Table 3: Fine-tuning state-of-the-art schedule baselines via grid searc: train loss (top) and eval loss (bottom). The base LR is tuned via a approximate multiplicative grid with a factor of 3. Warm-up and decay steps are tuned after the optimal base LR is found. N/A means the experiment fails and the loss blows up in the middle of training.

| LR | 3.33e−2 | 1e−2 | 3.33e−3 | 1e−3 | 3.33e−4 | 1e−4 | 3.33e−5 | 1e−5 |
|---|---|---|---|---|---|---|---|---|
| WSD | 6.889 | 3.460 | 5.253 | **3.184** | 3.635 | 4.691 | 5.415 | 6.129 |
| | 7.319 | 3.934 | 5.816 | **3.745** | 4.211 | 5.300 | 6.009 | 6.682 |
| Cosine | 6.477 | 3.887 | 4.626 | **3.417** | 4.019 | 5.092 | 5.869 | 6.681 |
| | 7.066 | 4.418 | 5.123 | **4.001** | 4.843 | 5.772 | 6.380 | 7.049 |
| WSD | N/A | N/A | N/A | N/A | **2.826** | 3.243 | 4.201 | 5.063 |
| (10k steps) | N/A | N/A | N/A | N/A | **3.261** | 3.701 | 4.744 | 5.681 |
| **Warm-up** | 0 | 100 | 200 | 500 | 1000 | | | |
| WSD | 3.308 | 3.273 | **3.184** | 3.327 | 3.349 | | | |
| | 3.845 | 3.744 | **3.745** | 3.788 | 3.852 | | | |
| Cosine | 3.665 | 3.402 | 3.417 | **3.465** | 3.465 | | | |
| | 4.229 | 4.020 | 4.001 | **3.995** | 3.997 | | | |
| Linear | 3.644 | 3.370 | **3.357** | 3.398 | 3.408 | | | |
| | 4.148 | 3.967 | **3.955** | 3.995 | 3.997 | | | |
| **Decay** | 0 | 100 | 200 | 500 | 1000 | | | |
| WSD | 3.168 | 3.251 | **3.184** | 3.218 | 3.207 | | | |
| | 3.841 | 3.759 | **3.745** | 3.754 | 3.808 | | | |

## B.2 FINE-TUNING OTHER SCHEDULES

Table 4: Tuning the base learning rate of the quadratic schedule and the triangle schedule: train loss (top) and eval loss (bottom).

| LR | 3.33e−2 | 1e−2 | 3.33e−3 | 1e−3 | 3.33e−4 | 1e−4 |
|---|---|---|---|---|---|---|
| Quadratic | 6.530 | 6.157 | **3.117** | 3.185 | 3.914 | 4.916 |
| | 7.234 | 6.715 | **3.632** | 3.836 | 4.547 | 5.519 |
| Triangle | 3.566 | **3.025** | 3.124 | 3.429 | 4.030 | 5.001 |
| | 7.028 | **3.652** | 3.678 | 3.976 | 4.850 | 5.719 |

## B.3 TUNING SEMI-LOCALITY CONSTANT

Table 5: Tuning semi-locality constant $\epsilon$ for semi-local search under different setups: train loss (top) and eval loss (bottom).

| $\epsilon$ | 0.0 | 0.06 | 0.12 | **0.24** | | | |
|---|---|---|---|---|---|---|---|
| 4 segs | 3.387 | 3.172 | 3.159 | **3.158** | | | |
| | 3.979 | 3.719 | 3.702 | **3.697** | | | |
| $\epsilon$ | 0.0 | 0.06 | **0.12** | 0.24 | 0.48 | | |
| 10 segs | 3.761 | 3.165 | **3.123** | 3.160 | 3.136 | | |
| | 4.401 | 3.711 | **3.653** | 3.683 | 3.658 | | |
| $\epsilon$ | 0.0 | 0.24 | **0.48** | 0.96 | | | |
| 10 segs (decay) | 3.761 | 3.112 | **3.093** | 3.132 | | | |
| | 4.401 | 3.646 | **3.612** | 3.652 | | | |
| $\epsilon$ | 0.0 | 0.015 | 0.03 | 0.06 | 0.12 | 0.24 | **0.48** |
| 20 segs | 4.869 | 4.346 | 4.190 | 4.309 | 4.918 | 5.143 | **3.653** |
| | 5.432 | 4.918 | 4.779 | 4.879 | 5.453 | 5.680 | **4.173** |

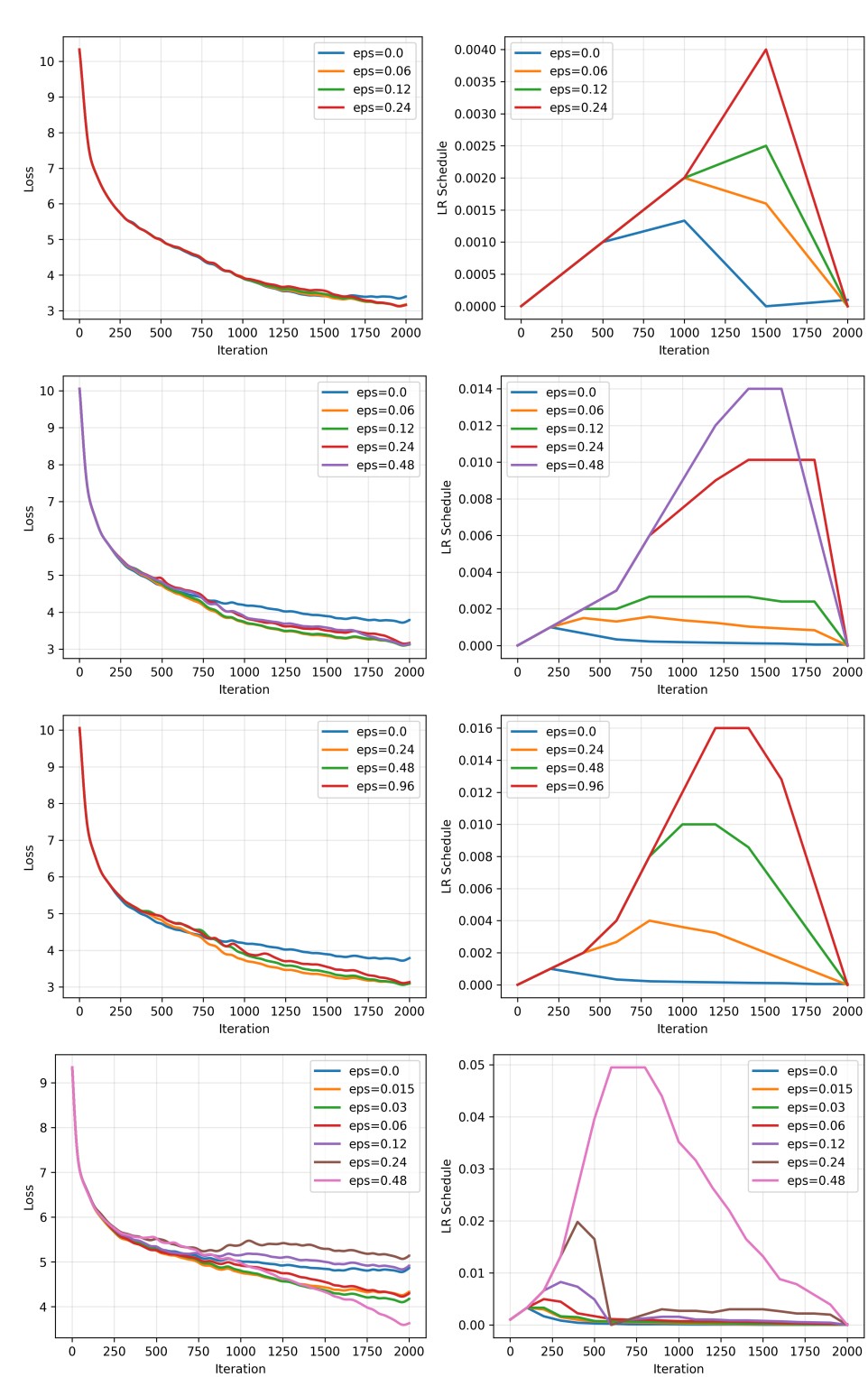

Figure 10: Semi-local search under 2000 train steps with different semi-locality constants: 4 segments (row 1), 10 segments (row 2), 10 segments with linearly decaying $\epsilon_k$ (row 3), and 20 segments (row 4). Left: train loss; Right: LR schedule.

