# OpenReview forum: "Semi-Local Search for LR Schedules"
_ICLR.cc/2026/Conference — Submitted to ICLR 2026_

### Official Review · Reviewer_TJFr · 2025-10-21

**Soundness:** 2
**Presentation:** 3
**Contribution:** 2
**Rating:** 4
**Confidence:** 2

**Summary:**

The authors investigate the use of semi-local search for learning rate scheduling, which move beyond standard local learning rate schedulers by selecting sub-optimal (but larger) learning rates at different timesteps. That is, the authors propose a method which splits up the training timesteps T, into K buckets, and within each bucket, the largest learning rate that is within epsilon of the optimal is selected. This avoids the types of greedy selection issues that local search might run into. In practice, the authors demonstrate improved loss compared to existing learning rate schedulers, and show that it can be approximated by a triangular or quadratic scheduler.

**Strengths:**

I will preface this by noting that this paper is far outside my usual range of expertise:
1. **Clear improvement over WSD and local search** - The authors demonstrate that the new proposed algorithm achieves lower loss against existing baselines, such as local search and WSD. For example, in Figure 4, the authors demonstrate that for appropriately chosen epsilon, the proposed algorithm outperforms WSD in loss, while local search is worse than WSD.
2. **Simple and intuitive algorithm** - The authors detail the proposed semi-local search in algorithm 1 and give a corresponding illustration in Figure 3. The figure nicely captures the way that their algorithm works in practice, by selecting the largest learning rate that is within the los tolerance. The algorithm is fairly intuitive, and the pseudocode is clean, making it easy for practitioners to implement in practice.

**Weaknesses:**

1. **Unclear whether this is generalizable** - It is unclear whether the results from this paper generalize across datasets or architectures. The authors evaluate on one dataset (more details on that below) and give little theoretical justification (more details on that as well below) for why this approach should work. In general, when proposing a new algorithm, one should either a) Thoroughly demonstrate the trend across different empirical settings, or b) Theoretically justify why this is the right approach. Instead, the arguments in Section 3 are largely based on pointing out some proposed issues with local search. It is unclear whether such issues persist across datasets/settings, and even if they do, it is unclear whether this would be the right fix.
2. **Only evaluated on a single dataset** - One disadvantage of this paper is that the proposed algorithm is only evaluated on a single dataset, namely MinGPT. While this is an interesting dataset (and one that, in principle, mimics GPT-style training), evaluation on one dataset limits the ability to judge whether the proposed learning rate scheduler performs well. This is especially important when proposing new learning rate schedulers because of their sensitivity to the dataset collected.
3. **Need for better justification** - In Section 3, the authors propose semi-local search because local search can perform worse than baselines such as WSD. However, it is unclear that the right solution to these issues is to select a larger learning rate. That is, there's little justification for why selecting within \epsilon of the optimal loss (per segment) improves performance. It would be nice to have some type of guarantee (e.g., if \epsilon is some value, then we can get the optimal loss under some assumptions), as this could at least provide intuition for why the method works and how to set epsilon.

**Questions:**

1. Why is it easier to tune and set \epsilon compared to setting learning rate?
2. Semi-local search seems to need to run lots of different learning rates in parallel per segment; is this feasible in practice?
3. Most of the results indicate a comparison with WSD; why was this selected as the main comparator?
4. In practice, is the suggestion to use some type of quadratic or triangle scheduler? Does semi-local always reproduce triangle?
5. It might be easier for presentation if all the figures/experiment-related things are kept in the experiment section. For example, the justification that local search performs poorly is justified through experiments, but then a new algorithm/method is presented in theory, making it confusing to jump back and forth.

---

### Official Review · Reviewer_CKx2 · 2025-10-29

**Soundness:** 1
**Presentation:** 2
**Contribution:** 1
**Rating:** 2
**Confidence:** 4

**Summary:**

This paper aims to replace classical learning-rate schedules with step sizes found by semi-local search. The main idea is to look ahead for a fixed amount of steps, and pick the largest schedule under a constraint of the achieved loss. Due to the allowed slack for the loss, the authors argue that the resulting schedule is not too small, which is the issue with fully local optimal schedules.

**Strengths:**

The paper tackles an important and practical problem, that is, how can we pick the best learning-rate schedule (semi-)automatically, or reduce the tuning cost for it.

**Weaknesses:**

* The proposed approach needs to run $n=10$ candidate schedules for each interval, and further needs to tune the value and decay of $\epsilon$. Therefore, the primary goal of this work, which is to reduce/automate the tuning effort, is not met in the end. The empirical evaluation can not convincingly show that the semi-local search indeed reduces tuning effort, as it does not report flop (or similar computational cost) comparisons in the main text (In the appendix some runtimes are reported though, but they are not in favor of the proposed method.). The paper further compares to baselines that are  -- in my opinion -- not carefully tuned (see more on this below), and therefore the reported performance gain might be equally well achieved by a much simpler approach: tune the known schedules (cosine, WSD) on a finer grid of peak learning rate (and potentially warmup length).

* The second main weakness of the experimental evaluation is clearly the extremely short training times. For 124M models, the experiments are restricted to 0.26B and 1.3B tokens. The chinchilla optimal length would be roughly 2.48B tokens, and in practice training runs are usually even longer than that. On top of this, the experiments show that the schedule found by semi-local search on a short horizon cannot be simply stretched to a long horizon (Figure 8); hence, the only convincing experiment would be to show that semi-local search achieves similar or better losses than carefully tuned cosine/WSD schedules on a long training run, with roughly the same computational budget. However, such an experiment is missing in the current submission.

* On the tuning of the baselines: the peak learning rate is tuned on a grid with factor 3.33. This is rather coarse compared to previous works on the same setup (e.g. see Figure 9 in https://arxiv.org/pdf/2509.01440). Further, warmup length, peak LR and cooldown length are not tuned jointly if I understood correctly. Due to this, the baseline WSD schedule is probably undertuned. One can also see this from Figure 4: eps=0.12 is essentially a WSD schedule with slightly longer warmup and higher peak LR, and it outperforms the baseline WSD schedule by quite a lot.

From the points mentioned above, one of the main conclusions, namely that *semi-local schedules outperforms state-of-the-art baselines* is not supported sufficiently by the experiments that are provided.

**Questions:**

* In Figure 1, the area under curve for the semi-local schedule is much bigger than for the baseline. In prior work it has been found that when carefully tuned, different schedules (multiplied by peak learning rate) usually have very similar area under curve. Can you explain why the semi-local approach would allow for a much larger area-under-curve?

* On section 4.3: this seems to be an interesting idea, but have you tested those schedule on realistic training lengths (at least 2.4B tokens)?

* Have you tested as a sanity check whether for nonsmooth, convex toy problems, the semi-local LR search results in linear-decay? It is known that linear-decay achieves the worst-case optimal convergence rate in that case, and usually also works best in practice. What is the resulting schedule of semi-local search with sufficient budget for toy problems?

* Given that the semi-local schedule results in rather high peak LRs, did you investigate whether this can cause loss instabilities for larger scale models?

Minor:

* The description of the tuning procedure in section 4.1 is hard to grasp from plain text. It might be helpful to add visualizations of the tuning results.

* The coloring of Figure 3 is confusing: what does orange/blue/gree mean, and why are the other segments in grey?

---

### Official Review · Reviewer_ZZ5r · 2025-11-01

**Soundness:** 2
**Presentation:** 3
**Contribution:** 2
**Rating:** 2
**Confidence:** 4

**Summary:**

The authors propose a "semi-local search" for finding an LR schedule online during training. "local search" works by, for each of K segments, running several different trials with different LRs and then continuing from the best one. Local search (and semi-local search) have the expressive power to find any piecewise linear schedules (for discrete choices of the LRs).

The delta from local search is that instead of choosing the LR with the *best* loss at the end of each segment, you choose the highest LR that is only epsilon worse than the best loss. This is a kind of exploration vs exploitation tradeoff, which has an intuitive appeal: We know we don't want to cooldown to early, which is what local-search would have us do.

The authors perform some very small scale experiments and then transfer to a somewhat less small experiment. The gains seem decently large but given the scale, the computational expense of the search, the lack of transfer, and the comparatively small amount of tuning of the baselines, I am not convinced there's anything here.

**Strengths:**

The paper is well written and the experiments are generally well explained. The basic idea seems appealing: allow the optimization to do more "exploration" (high LR, accepting higher losses) before tightening down later in training.

The authors are fairly exhaustive in their tuning, and make some (but in my opinion insufficient) effort ot tune the baseline.

I like that the authors tried to use the results of their non-parametric search to find parametric analogs, and seemingly succeed. I acutally think this is by far the best use case for semi local search: look for interesting patterns in the LR schedules with a high flexible nonparametric representation at small scale, then transfer to large scale. If the authors continue with this work, I encourage the authors to focus on that as I think it could be actually practical and impactful: use nonparametric search at small scale, transfer to larger scale. That should be the goal of this method, since running training 10 times isn't practical.

**Weaknesses:**

While I recognize the authors say they have limited resources, the reality is that the small scale and very short duration of the experiments makes the experiments pretty unconvincing. Most of the experiments were conducted on runs of up to 125M params/2000 steps, which is not a particularly relevant or realistic number of steps for pretty much any nontrivial modern problem, certainly not in the contexts where WSD would be used. Second, it has become conventional to use WSD with decays of as much as 20-40% of total steps. (e.g. kimi k2 https://arxiv.org/pdf/2507.20534, among others) 200 steps doesn't seem like a reasonable number.

Presumably part of the reason for the small scale of the experiments is just how incredibly expensive it is to do the search: for each time segment ,you run n trials (of length T/num_segments), meaning that you pay the price of training the model n times (n=number of learning rate candidates). They choose n=10, and sweep over epsilon and num_segments as well. (It's not clear how many epsilons they try, i think it's 8, but they try 3 different num_segments). This is a lot of tuning compared to the baseline!

For their main baseline WSD, they sweep 8 LRs and then, fixing the best LR, try 5 warmup durations, for a total of 13 runs. (They don't tune decay, which is probably more important.)

This could perhaps be okay if the schedule transferred to longer runs, but the best schedule from 2k steps scaled to 10,000 steps underperforms WSD (even after gridsearching this schedule) by as large of a margin as their semi-local searches that is tuned for that particular run outperform WSD. (And again, the semi-local search needs to be tuned over all possible learning rates (n=10 times), so they effectively tuned their algorithm 10x as much as WSD. The difference between WSD and semi-local at 10k steps is non-trivial, but seems like it could easily be due to undertuning of WSD.

**Questions:**

For WSD, I'd expect ~20-40% decay to be more useful in reality. 10% is very short and feels almost like a strawman. Can you do a sweep of 10,20,40% and ideally, if possible run the best at 10k steps.

---

### Official Review · Reviewer_Hufb · 2025-11-01

**Soundness:** 1
**Presentation:** 2
**Contribution:** 2
**Rating:** 2
**Confidence:** 5

**Summary:**

The paper proposes Semi-Local Search for automated on-the-fly learning-rate (LR) schedule discovery. Training is split into (K) segments; in each segment the method launches (n) candidate runs (linear schedules from current LR to candidates), collects segment losses, and selects the largest candidate whose loss is within (\epsilon_k) of the best (so it biases toward larger LRs while avoiding large loss increases).

**Strengths:**

Originality: The semi-local selection rule (choose the largest LR within an (\epsilon)-tube around the best loss) is a simple and intuitive operationalization of “bias towards larger LR while guarding loss” compared to purely greedy hyper-gradient or local methods.

Practical significance: If robust, the approach would offer a (first step towards) practical automated procedure for discovering schedules.

**Weaknesses:**

Unfortunately, the experiments in the paper are largely insufficient. This is not only because of a very small scale but, in particular, extremely short runs. I understand that there are restrictions to experiments on an academic budget, but 2000 or 10000 steps for a single model size and setup do not validate the findings rigorously.
I additionally have doubts about the validity, e.g., since there should not be such a large gap between cosine and WSD if both are well tuned; perhaps the final LR are not set properly. Also, why tune the base LR first and then tune the warmup steps? The opposite is the natural order. Moreover, very natural and necessary ablations are missing, such as investigations into runtime overhead. The only analyses are loss vs. steps and final losses in tables.

Overall, the quality of experiments and the paper unfortunately do not meet the bar.

**Questions:**

If on a tight academic budget, I would strongly suggest other experimental setups (e.g., ImageNet, ResNets, ...) to validate methods first before moving to costly language models.

---

### Meta-Review · Area_Chair_fWCK · 2026-01-06

**Summary:**

The paper proposes "Semi-Local Search," an automated method for discovering effective learning rate (LR) schedules on the fly. The authors argue that standard "local search" performs poorly because it favors small LRs, resulting in slower convergence. To address this, the proposed algorithm selects the largest LR whose loss is within a small margin of the best-performing candidate (instead of greedily picking the LR with the lowest loss). This "semi-local" bias encourages exploration and larger step sizes while maintaining stability. Experiments on minGPT demonstrate that semi-local search can outperform tuned baselines such as WSD and Cosine.

**Reviewer Concerns:**

The reviewers consistently pointed out some experimental limitations, including:
1. The experiments were restricted to small-scale models.
2. The training durations were extremely short.
3. The search process incurs a significant computational overhead.
4. The baselines appear to be undertuned or configured unconventionally.
5. The method was evaluated on a single specific setup: minGPT trained on the Pile dataset.

Unfortunately, the authors were not involved in the rebuttal process, so these concerns remain unaddressed.

**Reviewer Scores:**

Given that these critical concerns remain unaddressed, the AC believes the reviewers would maintain their negative scores had they been able to participate fully in the discussion.

---

### Decision · Program_Chairs · 2026-01-26

Reject